# Radiomic features of axillary lymph nodes based on pharmacokinetic modeling DCE-MRI allow preoperative diagnosis of their metastatic status in breast cancer

Hong-Bing Luo[1ᵒ], Yuan-Yuan Liu[1ᵒ], Chun-hua Wang[1], Hao-Miao Qing[1], Min Wang[1], Xin Zhang[2], Xiao-Yu Chen[1], Guo-Hui Xu[1], Peng Zhou[1]*, Jing Ren[1]*

1 Department of Radiology, Sichuan Cancer Hospital & Institute, School of Medicine, University of Electronic Science and Technology of China, Chengdu, China, 2 Pharmaceutical Diagnostic Team, GE Healthcare, Life Sciences, Beijing, China

ᵒ These authors contributed equally to this work.
* 13880611648@163.com (JR); penghyzhou@126.com (PZ)

**Data Availability Statement:** Our Medical Ethics Committee imposed ethical and legal restrictions on sharing a de-identified data set, because the

## Abstract

### Objective

To study the feasibility of use of radiomic features extracted from axillary lymph nodes for diagnosis of their metastatic status in patients with breast cancer.

### Materials and methods

A total of 176 axillary lymph nodes of patients with breast cancer, consisting of 87 metastatic axillary lymph nodes (ALNM) and 89 negative axillary lymph nodes proven by surgery, were retrospectively reviewed from the database of our cancer center. For each selected axillary lymph node, 106 radiomic features based on preoperative pharmacokinetic modeling dynamic contrast enhanced magnetic resonance imaging (PK-DCE-MRI) and 5 conventional image features were obtained. The least absolute shrinkage and selection operator (LASSO) regression was used to select useful radiomic features. Logistic regression was used to develop diagnostic models for ALNM. *Delong* test was used to compare the diagnostic performance of different models.

### Results

The 106 radiomic features were reduced to 4 ALNM diagnosis–related features by LASSO. Four diagnostic models including conventional model, pharmacokinetic model, radiomic model, and a combined model (integrating the Rad-score in the radiomic model with the conventional image features) were developed and validated. *Delong* test showed that the combined model had the best diagnostic performance: area under the curve (AUC), 0.972 (95% CI [0.947–0.997]) in the training cohort and 0.979 (95% CI [0.952–1]) in the validation cohort. The diagnostic performance of the combined model and the radiomic model were better than that of pharmacokinetic model and conventional model (P<0.05).

data contain potentially identifying and sensitive patient information. Upon request, the request for data should be sent to Our Medical Ethics Committee (Email:scchec@163.com) or to the corresponding author (Jing Ren, Email: 13880611648@163.com).

**Funding:** This work was supported by Project of Sichuan Medical Association (www.sma.org.cn) Grant# S17067 (Y.Y. L),Sichuan Science and Technology Program (http://kjt.sc.gov.cn/) Grant# 2021YFS0075 (J. R), and Sichuan Science and Technology Program(http://kjt.sc.gov.cn/) Grant# 2021YFS0225 (P. Z). The funders had no role in study design, data collection and analysis, decision to publish, or preparation of the manuscript. XZ received support in the form of a salary from GE Healthcare. The specific roles of these authors are articulated in the 'author contributions' section. The funders had no role in study design, data collection and analysis, decision to publish, or preparation of the manuscript.

**Competing interests:** The authors of this manuscript have read the journal's policy and have the following competing interests: XZ is an employee of GE Healthcare. There are no patents, products in development or marketed products associated with this research to declare. This does not alter our adherence to PLOS ONE policies on sharing data and materials.

**Abbreviations:** ALNM, Metastatic axillary lymph nodes; ALNC, Axillary lymph nodes for control; PK-DCE-MRI, Pharmacokinetic modeling dynamic contrast-enhanced magnetic resonance imaging; LASSO, Least absolute shrinkage and selection operator; ROC, Receiver operating characteristic; AUC, Area under the curve; FOV, Field of view; TA, Total acquisition time; CAIPIRINHA-Dixon-TWIST-VIBE, Controlled aliasing in parallel imaging results in higher acceleration-Dixon-Time resolved imaging with interleaved stochastic trajectories-Volume interpolated body examination; ROI, Region of interest; VOI, Volume of interest; DWI, Diffusion weighted imaging.

## Conclusion

Radiomic features extracted from PK-DCE-MRI images of axillary lymph nodes showed promising application for diagnosis of ALNM in patients with breast cancer.

## Introduction

Breast cancer is the most common malignant cancer in women worldwide as well as in China, and has a high mortality rate [1, 2]. Lymphatic metastasis is the first step in the transition of breast cancer patients to metastatic state, and axillary lymphatic node metastasis is an important predictor of breast cancer recurrence [3]. Therefore, pre-treatment diagnosis of metastatic axillary lymph node (ALNM) is crucial for prognostic assessment and treatment decision-making [4–7]. Currently, lymphadenectomy and/or biopsy is the gold standard for differentiating ALNM from normal lymph nodes; however, these are invasive procedures associated with low repeatability and potential complications [5, 8]. Therefore, development of alternative noninvasive and repeatable methods for preoperative identification of ALNM is a key imperative.

Dynamic contrast-enhanced magnetic resonance imaging (DCE-MRI) is widely used for preoperative evaluation of axillary lymph node status in patients with breast cancer and shows superior performance than other techniques [9, 10]. The traditional DCE-MRI diagnostic criteria for ALNM are based on visual assessment of morphological features [11]. Radiomics provides an innovative quantitative method to predict ALNM in patients with breast cancer [7, 12–19].

Radiomics refers to the science of converting medical images to high-throughput and mineable quantitative features by data characterisation algorithms [20]. These features, termed radiomic features, have the potential to decode the invisible disease characteristics, which are useful for individualized treatment. It is different from the traditional medical images, which are subject to visual interpretation. Use of modern analytical software and artificial intelligence technology has helped unravel an increasing number of useful features obtained through radiomic method, especially in the field of cancer research [21].

Breast cancer is known to be a heterogeneous disease caused by variations in local microenvironment that are mainly governed by spatial and temporal changes in blood flow. Tumor heterogeneity may be represented by different contrast-enhancement patterns on DCE-MRI and amenable to quantitative assessment using radiomic methods based on PK-DCE-MRI [22–25]. Some recent studies have shown that radiomic features extracted from primary breast cancer mass may be used to predict metastases in the sentinel lymph node [15, 16] and axillary lymph nodes [18, 19]. However, to the best of our knowledge, the role of radiomic features extracted from axillary lymph nodes for diagnosis of their metastatic status is still not studied.

The ALNM of breast cancer, which is similar to the tumor itself, may also exhibit heterogeneity; the heterogeneous characteristics on PK-DCE-MRI can be decoded by radiomic method. Our hypothesis is that the radiomic features extracted from PK-DCE-MRI images of axillary lymph nodes can help diagnose ALNM in patients with breast cancer.

## Materials and methods

### Subjects

The Medical Ethics Committee of the Sichuan Cancer Hospital & Institute approved this study. The requirement for informed consent of subjects was waived off. We retrospectively

reviewed our database to select consecutive women with proven breast cancer by surgical pathology and who underwent DCE-MRI examination before surgery between August 2015 and June 2019. The exclusion criteria were: patients who had received neo-adjuvant chemotherapy; patients for whom quantitative parameters could not be acquired due to data-processing errors.

To ensure that axillary lymph nodes included in radiomic analyses were pathologically metastatic nodes, cases recruited to the ALNM group were required to qualify the following two conditions. First, at least 3 metastatic axillary lymph nodes were confirmed by pathology after axillary lymphadenectomy. Second, there was at least one highly suspicious axillary lymph node on DCE-MRI images in the ipsilateral axilla, which was visible to radiologists. Finally, two radiologists with 8 and 6 years of experience in the interpretation of breast MRI, respectively, reviewed the surgical pathology reports and MRI images together, and selected only one largest and highly suspicious axillary lymph node for each recruited patient for radiomic analysis.

To ensure that axillary lymph nodes included in radiomic analyses were pathologically negative for metastasis, cases recruited to the negative axillary lymph nodes group for control (ALNC) were required to qualify the following conditions. First, all axillary lymph nodes of the patients with negative sentinel lymph node biopsy were considered negative [13]. Second, we only chose the largest visible ipsilateral axillary lymph node of these patients for radiomic analysis.

## MRI acquisition

The MRI acquisition (as briefly described below) in this study were not specific to the current research and have been described extensively in our previous study [26]. All DCE-MRI examinations were performed using a 3.0-T Skyra device (Siemens Healthcare, Erlangen, Germany) with a dedicated breast coil (16-channel breast array; Siemens Healthcare, Erlangen, Germany). With the patient in a prone position, Axial T2-weighted imaging (T2WI), diffusion-weighted imaging (DWI) and DCE-MRI sequences were obtained. The DCE-MRI included T1 mapping and 26 consecutive phases fast dynamic MR acquisition, using the CAIPIRIN-HA-Dixon-TWIST-VIBE sequence with the temporal resolution 11.8 s/phase and TA 5 min 12 s. Gadodiamide (0.1 mmol/kg; Omniscan, GE Healthcare, Milwaukee, WI) was intravenously administered using a power injector (rate, 2.5 mL/s) at the end of T1 mapping. Then, a 20-mL saline flush was injected (rate, 2.5 mL/s).

## Post-processing of MRI images and radiomic analysis

Raw DCE-MRI data were imported into a dedicated post-processing software (Omni-Kinetics, GE Healthcare, Milwaukee, WI). The enhancement kinetics were analyzed using the reference region model [27]. With the reference region set to pectoralis major muscle, voxel-wise perfusion maps were automatically generated.

A schematic illustration of the radiomic analysis process is shown in Fig 1; the process consisted of 3D whole lymph node segmentation, features extraction, features selection, model building, and evaluation.

**Whole lymph node segmentation.** For extracting the radiomic features of each selected lymph node, the same two radiologists manually drew the regions of interest (ROIs) that included the whole lymph node in the early stage of postcontrast image of DCE-MRI. The radiologist with 8 years of experience performed all definitive measurements. Subsequently, all ROIs were merged into one 3D volume of interest (VOI).

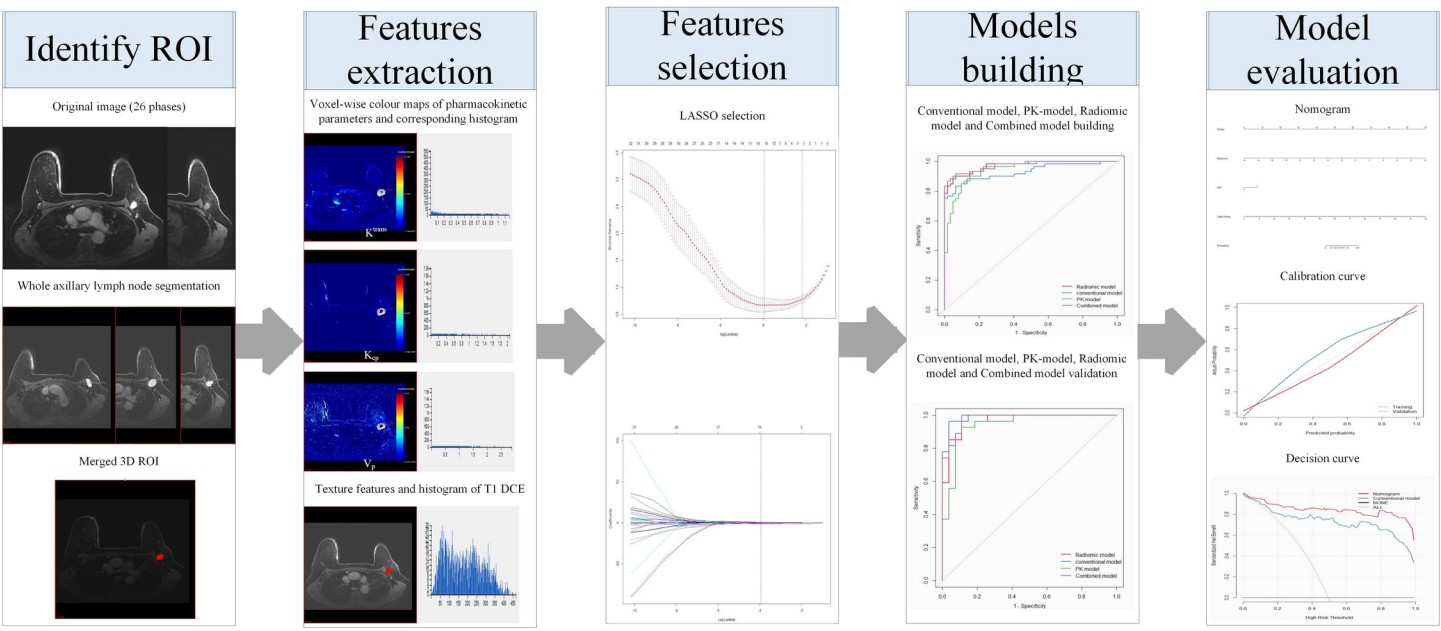

**Fig 1. General radiomic workflow in the study.**

**Features extraction.** A total of 106 radiomic features were extracted and automatically outputted by the software from the VOI, including 1 total voxel number of VOI, 30 standard pharmacokinetic parameters with their corresponding histogram features: (1) $K^{trans}$ ($min^{-1}$), the volume-transfer constant (wash-in rate), which reflects vascular permeability and perfusion; (2) $K_{ep}$ ($min^{-1}$), the washout-rate constant, which reflects contrast-agent reflux back to the vessels; (3) $V_p$, the plasma fraction. The histogram features, i.e., the maximum, minimum, median, mean, Std, 10%, 25%, 50%, 75%, and 90% values, of each quantitative parameter, and 75 texture features of T1-weighted images of dynamic contrast enhanced-magnetic resonance imaging (T1 DCE-MRI) (S1 Table), consisting of 12 first-order statistical features, 15 histogram features, 13 Gray-level co-occurrence matrix (GLCM) features, 10 Haralick features, 16 Run length matrix (RLM) features and 9 Morphology metrics features.

**Features selection and radiomic model building.** First, the training and validation cases were separated at a ratio of 7 to 3. We employed the least absolute shrinkage and selection operator (LASSO) technique and leave-one-out cross-validation (LOOCV) method to select and rank the optimal radiomic features from the primary date set in the training cohort. Tunning minimum criteria selection in the LASSO model used 10-fold cross validation via minimum criteria in the study. Then we used logistic regression to build a pharmacokinetic model (PK-model) based on pharmacokinetic parameters with their corresponding histogram features and a radiomic model based on radiomic features. The Radiomics score (Rad-score) was calculated for each patient according the coefficients of the radiomic model, which was defined as a radiomic signature. The performance was then validated in the validation cohort.

**Conventional model building.** For conventional model building, some conventional image features of every selected node (including the long and short axis length, short-long axis ratio, fatty hilum status on T2WI sequence, signal intensity on diffusion weighted imaging (DWI), and heterogenous enhancement feature on DCE-MRI sequences) were assessed by the same two radiologists. We used univariate analyses to compare these features between ALNM and ALNC (Table 1). We used logistic regression to build a conventional model based on these

**Table 1. Difference of conventional image features between ALNM and ALNC.**

| Features | | ALNC (N = 89) | | ALNM (N = 87) | | p value |
|---|---|---|---|---|---|---|
| long axis length (mm, mean±SD) | | 8.73 | ±3.15 | 17.92 | ±10.43 | 0.000 |
| short axis length (mm, mean±SD) | | 5.23 | ±1.98 | 12.30 | ±8.34 | 0.000 |
| short-long axis ratio | | 1.78 | ±0.61 | 1.52 | ±0.36 | 0.001 |
| fatty hilum status | no | 39 | | 71 | | 0.000 |
| | yes | 50 | | 16 | | |
| signal intensity on DWI | low | 18 | | 6 | | 0.010 |
| | high | 71 | | 81 | | |
| heterogenous enhancement feature | no | 76 | | 19 | | 0.000 |
| | yes | 13 | | 68 | | |

Note: ALNC, Axillary lymph nodes for control; ALNM, Metastatic axillary lymph nodes; DWI, Diffusion weighted imaging

candidate features with p < 0.05 in the univariate analyses. The performance was then validated in the validation cohort.

**Combined model building.** Integrating the Rad-score in the radiomic model with the conventional image features, the combined model was built using the multivariable logistic regression method. The performance of the combined model was validated in the validation cohort.

**Comparison of models.** Receiver operating characteristic (ROC) curve analysis was used to evaluate the diagnostic performance of the above models in diagnosing ALNM. The *Delong* test was used to compare the diagnostic performance of the conventional model, radiomic model, and the combined model according to the area under the curve (AUC) values.

**Establishment of nomogram.** To provide an individualized tool for ALNM diagnosis, a nomogram based on the combined model was plotted. The calibration of the nomogram was assessed using calibration curve, accompanied with the Hosmer–Lemeshow test. Harrell's C-index was measured to quantify the discriminative ability of the nomogram.

**Clinical use.** Decision curve analysis was conducted to determine the clinical usefulness of the nomogram by quantifying the net benefits at different threshold probabilities in the validation dataset.

## Statistical analysis

R statistical software (version 3.6.1) was used for statistical analyses. P values < 0.05 were considered indicative of statistical significance.

## Results

### Patients and axillary nodes in the study

Finally, a total of 176 axillary lymph nodes (87 ALNM breast cancer patients [mean age, 50.7 years; range, 28–78] and 89 ALNC breast cancer patients [mean age, 50.0 years; range, 30–76] were selected for radiomic analyses.

### Radiomic features selection and radiomic signature building

The 106 radiomic features of each selected axillary lymph node were reduced to 4 ALNM diagnosis–related features by LASSO (Fig 2). They were all texture features of T1 DCE-MRI, including Haralick Correlation, Difference Variance, DifferenceEntropy and LongRunEmphasis. A radiomic signature containing these features was constructed. The diagnostic

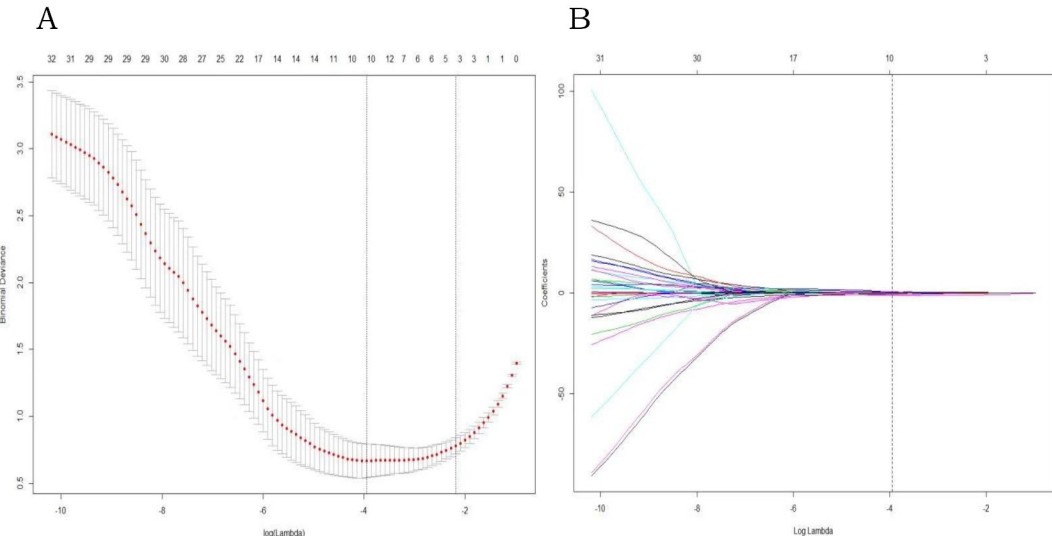

**Fig 2. LASSO regression for radiomic feature selection.** (A) Selection of the parameter (λ) in the LASSO model by 10-fold cross-validation based on minimum criteria. The y-axis indicates binomial deviances. The lower x-axis indicates the log(λ). Red dots indicate the average deviance values for each model with a given λ, and vertical bars through the red dots show the upper and lower values of the deviances. The dotted vertical lines define the optimal values of λ, where the model provides the best fit to the data. (B) A coefficient profile plot was produced against the log (l) sequence. Vertical line was drawn at the value selected using 10-fold cross-validation, where optimal λ resulted in non-zero coefficients.

performance of the radiomic signature was excellent. The optimal cutoff value of 0.38 was associated with an AUC of 0.971 (95% CI [0.947–0.995]) in the training cohort and 0.966 (95% CI [0.925–1]) in the validation cohort. The accuracy in the training and validation cohorts was 91.8% and 90.7%, respectively. The sensitivity of the radiomic signature in the training and validation cohorts was 90% and 92.6%, respectively; the specificity was good (93.5% in the training cohort and 88.9% in the validation cohort.

## Building of diagnostic models

The conventional model based on conventional image features, PK-model based on pharmacokinetic parameters with their corresponding histogram features, radiomic model based on all radiomic features, and the combined model integrating radiomic features with conventional image features were built. The diagnostic performance of these models is shown in Table 2. The combined model showed the best diagnostic performance; the optimal cutoff value of 0.671 was associated with an AUC of 0.972 (95% CI [0.947–0.997]) in the training cohort and 0.979 (95% CI [0.952–1]) in the validation cohort. The accuracy in the training and validation cohorts was 92.6% and 92.6%, respectively. The sensitivity of the radiomic signature in the training and validation cohorts was 91.7% and 96.3%, respectively; the specificity was good (93.5% in the training cohort and 88.9% in the validation cohort).

## Comparison of models and development of nomogram

The diagnostic performance of combined model and radiomic model was better than that of the conventional model (*Delong* test, p<0.05). However, the diagnostic performance of the combined model was not better than that of the radiomic model (p>0.05). The ROC curves of the four models are shown in Fig 3. We developed a nomogram based on the radiomic signature and conventional image features (Fig 4). The calibration curve for the nomogram was

**Table 2. Diagnostic performance of all models for detection of metastatic axillary lymph node.**

| | Training | | | | Validation | | | |
|---|---|---|---|---|---|---|---|---|
| | AUC (95% confidence interval) | ACC | Specificity | Sensitivity | AUC (95% confidence interval) | ACC | Specificity | Sensitivity |
| Radiomic model | 0.971 (0.947–0.995) | 0.918 | 0.935 | 0.9 | 0.966 (0.925–1) | 0.907 | 0.889 | 0.926 |
| Conventional model | 0.929 (0.881–0.977) | 0.877 | 0.871 | 0.883 | 0.988 (0.968–1) | 0.926 | 0.889 | 0.963 |
| PK model | 0.945 (0.908–0.981) | 0.877 | 0.855 | 0.9 | 0.942 (0.88–1) | 0.87 | 0.815 | 0.926 |
| Combined model | 0.972 (0.947–0.997) | 0.926 | 0.935 | 0.917 | 0.979 (0.952–1) | 0.926 | 0.889 | 0.963 |

Note: AUC, area under the curve; ACC, accuracy; PK-model, pharmacokinetic model.

tested by Hosmer-Lemeshow test, which yielded a non-significant result ($\chi^2 = 3$, p>0.05) showing good calibration (Fig 5).

## Clinical use

The decision curve analyses for the nomogram and conventional model are presented in Fig 6. The results showed that the net benefit of using the radiomics nomogram for diagnosis of ALNM was greater than that of the conventional model.

## Discussion

The results of this study, though preliminary in scope, reveal that the radiomic features extracted from preoperative PK-DCE-MRI of axillary lymph nodes can be used to diagnose ALNM in patients with breast cancer. In addition, in this study, the texture features depicting heterogeneity were found more helpful than pharmacokinetic quantitative and their histogram features for diagnosis of metastatic axillary nodes.

Use of radiomics analysis for diagnosis and prognostic assessment in the context of breast cancer is a contemporary research hotspot. Most of the studies have focused on discriminating malignant from benign breast tumors [28–30] or on predicting the chemotherapeutic response

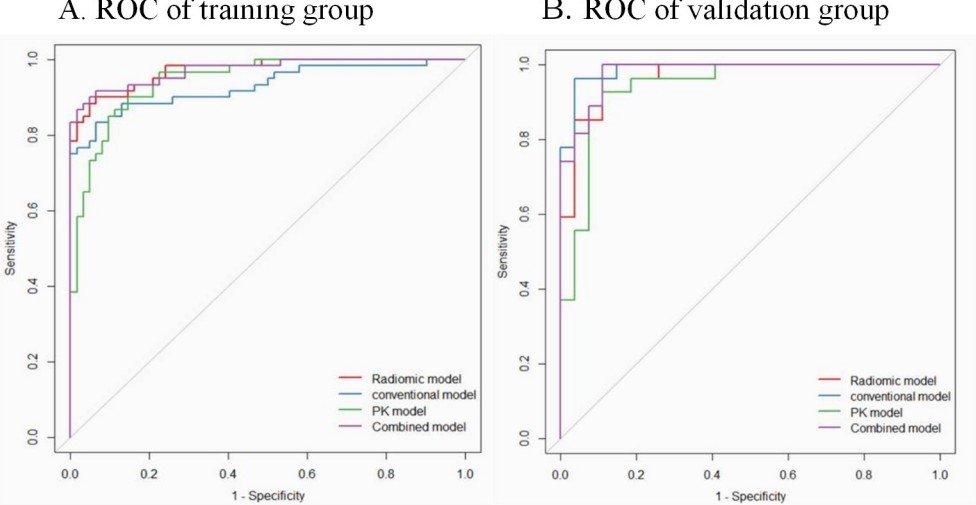

**Fig 3.** Receiver operating characteristic (ROC) curves of conventional model, pharmacokinetic model (PK-model), radiomic model, and the combined model for diagnosis of metastatic axillary lymph nodes in the training (A) and validation (B) group at a ratio of 7 to 3.

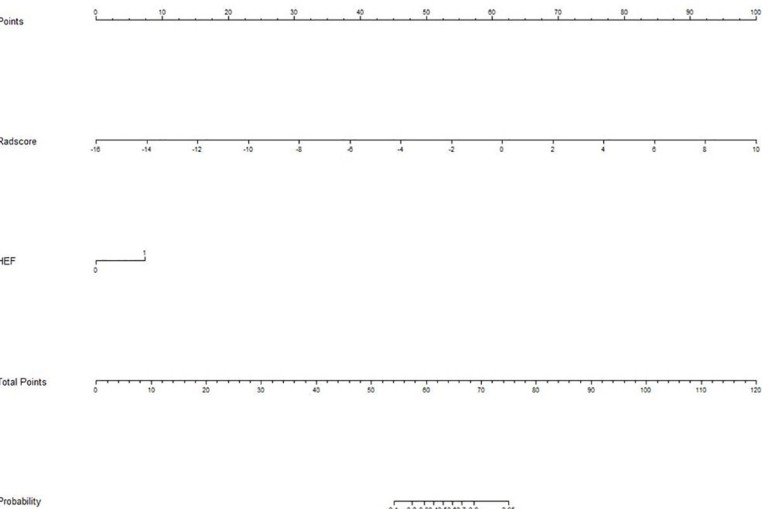

**Fig 4. Nomogram for diagnosis of metastatic axillary nodes.** The values for each variable correspond to a point at the top of the graph, and the sum of the points for all the variables corresponds to a total point; a line drawn from the total points to the bottom line shows the probability of axillary lymph nodes metastasis. Heterogenous enhancement feature (HEF) is a conventional image feature.

[31, 32]. Some recent studies have investigated the feasibility of differentiating ALNM by radiomic features, most of these studies were focused on the predictive value of radiomic features extracted from primary breast tumors [17–19]. The radiomic features of breast tumors

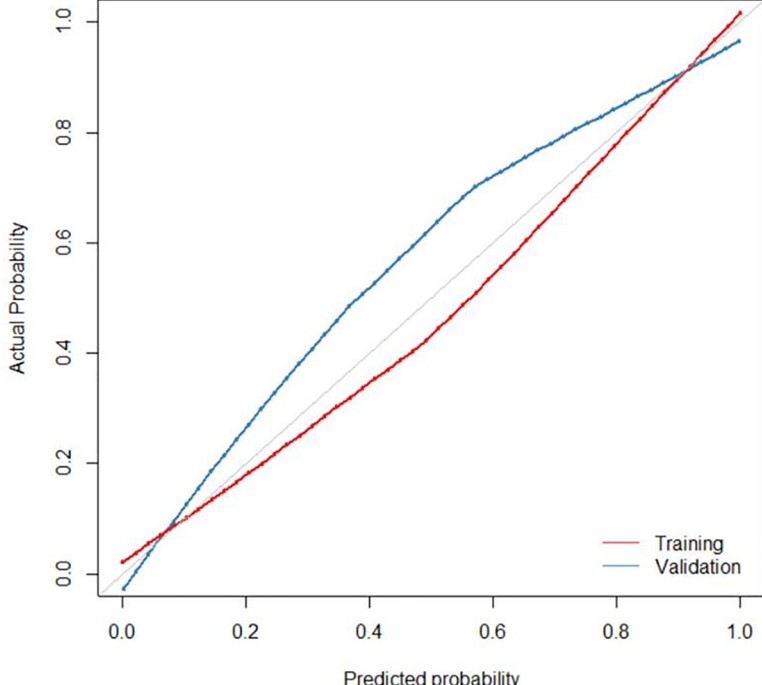

**Fig 5. Calibration curve of the nomogram for the training (red) and validation (blue) cohorts at a ratio of 7 to 3.** The X-axis represents the probability that nomogram diagnosed the axillary lymph nodes metastasis, while Y-axis represents the actual rate of axillary lymph nodes metastasis.

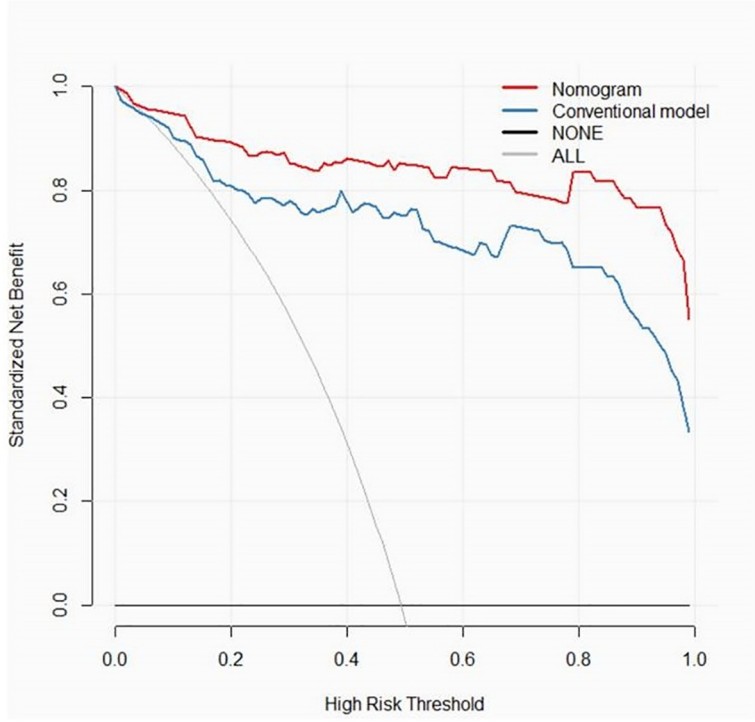

**Fig 6. Decision curve analysis of the nomogram of combined model and conventional model.** The y-axis measures the net benefit. The red line represents the nomogram of the combined model. The blue line represents the conventional model. The thin grey line represents the assumption that all patients have axillary lymph nodes metastases. The black line represents the assumption that none of the patients have axillary lymph nodes metastases.

may be used to predict the ALNM; however, these can not be directly used to diagnose ALNM. Therefore, we designed this study to assess the feasibility of use of radiomic features extracted from axillary lymph nodes for diagnosis of their metastasis status. Our results show that the diagnostic performance of radiomic model for detecting ALNM was better than that of the conventional model. Although not better than radiomic model, the combined model showed the best diagnostic performance among the four models. The accuracy was improved from 0.877 of conventional model to 0.926 of the combined model. These results implied that the radiomic features of axillary lymph node itself are promising bio-markers for helping diagnosing its metastasis status. Besides, all the 4 top-rank radiomic features for discrimination between positive and negative axillary nodes by LASSO regression were texture features of T1 DCE-MRI. This suggests that the texture features have a more prominent discriminative performance than the pharmacokinetic parameters and their corresponding histogram features. These findings are consistent with those a previous study by Schacht et al in which kinetic features showed poorer performance in distinguishing between positive and negative lymph nodes [13]; this was attributable to the fact that some normal axillary lymph nodes may also exhibit patterns of rapid uptake and washout kinetics. The results implied that heterogeneity may be the most important characteristic of ALNM in breast cancer, which is difficult to interpret on visual examination of conventional images; however, it can be depicted by radiomic methods through whole-tumor and voxel-wise quantitative analysis based on PK-DCE-MRI. This is concordant with our hypothesis that ALNM are spatially more heterogeneous than the negative ones. The heterogeneous characteristics of axillary lymph nodes could be comprehensively decoded through radiomic methods on the basis of PK-DCE-MRI *in vivo*; this could be

used as an effective supplement to traditional medical images for diagnosis of ALNM of breast cancer in future.

This study had certain limitations. First, the effect of selection bias on our results cannot be ruled out, as it is difficult to accurately match the selected axillary lymph nodes for radiomic analyses visible on DCE-MRI with the metastatic nodes proven by resection and biopsy. To minimize this bias, we only recruited patients for whom at least 3 axillary lymph nodes metastases were confirmed by pathology after axillary lymphadenectomy; subsequently, we selected only one of the most likely axillary lymph nodes in the ipsilateral axilla as ALNM. Second, the semi-automatic features extraction approach may cause some inter-observer variability. With advances in software and algorithms, the detection and segmentation method should be combined with computer vision algorithms for automated specification of the VOI beyond human perception. Such approaches may theoretically be easily implemented in clinical workflow. Finally, although our results were based on high-field strength MRI and the CDT-VIBE sequence protocol, which has high temporal and spatial resolution essential for PK-DCE-MRI, this was only a single-center study. Thus, our findings need to be verified in a multi-center study with different imaging equipments and protocols.

## Conclusions

Based on the preliminary results of our study, it may be inferred that the ALNM of breast cancer are more heterogeneous than the negative nodes. Radiomic methods can be used to decode the heterogeneity of axillary lymph nodes. Our findings provide impetus for further radiomic research to develop a non-invasive tool for diagnosis of metastatic lymph node and individualized treatment.

## Supporting information

**S1 Table. Texture features of T1 DCE-MRI in the study.**
(DOC)

## Author Contributions

**Conceptualization:** Hong-Bing Luo, Yuan-Yuan Liu, Chun-hua Wang, Hao-Miao Qing, Peng Zhou, Jing Ren.

**Data curation:** Hong-Bing Luo, Yuan-Yuan Liu, Chun-hua Wang, Hao-Miao Qing, Min Wang, Peng Zhou, Jing Ren.

**Formal analysis:** Hong-Bing Luo, Yuan-Yuan Liu, Hao-Miao Qing, Xin Zhang, Peng Zhou, Jing Ren.

**Funding acquisition:** Yuan-Yuan Liu, Peng Zhou, Jing Ren.

**Investigation:** Hong-Bing Luo, Yuan-Yuan Liu, Chun-hua Wang, Min Wang, Xiao-Yu Chen, Jing Ren.

**Methodology:** Hong-Bing Luo, Yuan-Yuan Liu, Chun-hua Wang, Hao-Miao Qing, Min Wang, Xiao-Yu Chen, Jing Ren.

**Project administration:** Hong-Bing Luo, Yuan-Yuan Liu, Guo-Hui Xu, Peng Zhou, Jing Ren.

**Resources:** Hong-Bing Luo, Yuan-Yuan Liu, Peng Zhou, Jing Ren.

**Software:** Hong-Bing Luo, Yuan-Yuan Liu, Hao-Miao Qing, Xin Zhang, Xiao-Yu Chen, Peng Zhou, Jing Ren.

**Supervision:** Hong-Bing Luo, Min Wang, Guo-Hui Xu, Peng Zhou, Jing Ren.

**Validation:** Hong-Bing Luo, Yuan-Yuan Liu, Chun-hua Wang, Jing Ren.

**Visualization:** Hong-Bing Luo, Yuan-Yuan Liu, Peng Zhou, Jing Ren.

**Writing – original draft:** Hong-Bing Luo.

**Writing – review & editing:** Hong-Bing Luo, Yuan-Yuan Liu, Jing Ren.

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
