## [Decision Letter · Decision Letter 0]

17 Dec 2020

PONE-D-20-30485

Radiomic features of axillary lymph nodes based on pharmacokinetic modeling DCE-MRI allow preoperative diagnosis of their metastatic status in breast cancer

PLOS ONE

Dear Dr. Ren,

Thank you for submitting your manuscript to PLOS ONE. After careful consideration, we feel that it has merit but does not fully meet PLOS ONE’s publication criteria as it currently stands. Therefore, we invite you to submit a revised version of the manuscript that addresses the points raised during the review process.

Performing a careful comparison with the conventional image reading of the nodes to demonstrate how the radiomics model may help in the cases.Questions about tables.

We look forward to receiving your revised manuscript.

Kind regards,

Quan Jiang, Ph,D.

Academic Editor

PLOS ONE

Journal Requirements:

4. Thank you for stating the following financial disclosure: 'No'

5. Thank you for stating the following in the Competing Interests section: 'No' 

We note that one or more of the authors are employed by a commercial company, GE Healthcare.

6. Please include your tables as part of your main manuscript and remove the individual files. Please note that supplementary tables should be uploaded as separate "supporting information" files.

Reviewers' comments:

Reviewer's Responses to Questions

**Comments to the Author**

1. Is the manuscript technically sound, and do the data support the conclusions?

Reviewer #1: Yes

Reviewer #2: Yes

2. Has the statistical analysis been performed appropriately and rigorously? 

Reviewer #1: Yes

Reviewer #2: Yes

3. Have the authors made all data underlying the findings in their manuscript fully available?

Reviewer #1: Yes

Reviewer #2: Yes

4. Is the manuscript presented in an intelligible fashion and written in standard English?

Reviewer #1: Yes

Reviewer #2: Yes

5. Review Comments to the Author

Reviewer #1: This is a well written study to report the differentiation of 87 malignant and 89 benign axially lymph nodes seen in breast MRI, by using 4 different models: 1) conventional model based on image features; 2) PK-model based on pharmacokinetic parameters with their corresponding histogram features; 3) radiomic model based on all radiomic features, and 4) the combined model integrating radiomic features with conventional image features. The methods were pretty standard and well described. The results showed that the radiomics model based on texture had the best performance, and that was attributed to the heterogeneity in malignant compared to benign nodes. Although the results could achieve very high accuracy, it is still difficult to implement the complicate models in a clinical setting. Performing a careful comparison with the conventional image reading of the nodes to demonstrate how the radiomics model may help in the cases that the reading was wrong will greatly improve the value of this work. Some comments below:

1. In the manuscript that I downloaded for review, Tables 1 and 2 were not included, thus it was difficult to compare the performance (sensitivity vs. specificity) between the radiomics and conventional models, which is the most interesting part in my opinion.

2. There are well established nodal features commonly used to predict malignant vs. benign nodes, as described in this work, including the long and short axis length, short-long axis ratio, fatty hilum status, signal intensity on diffusion weighted imaging (DWI), and heterogenous enhancement feature. Please add a table to compare these features between the malignant and benign groups.

3. Please give the number of true positive, true negative, false positive, and false negative cases predicted by each of the 4 models. Then compare the results of reading and radiomics to see which cases were correctly predicted by radiomics but wrongly diagnosed by reading. Show these case examples using figures. These results will be very interesting.

4. The presence of fatty hilum is considered as an important benign feature, but it is not clear how this feature can be well appreciated on the DCE imaging sequence used in this study, that heavily focuses on a high temporal resolution (thus may sacrifice the spatial resolution). What other MR imaging sequences were considered in the reading? Please discuss.

5. After adding results in issues 2-4, please also add discussion to elaborate the value of the radiomics that may be complementary to reading. In fact, it seems all 4 models have very high accuracy; therefore, the actual value of the very complicated models may be limited. Please emphasize on the discordant cases between reading and radiomics models.

6. Which DCE frame was used for drawing the nodal ROI? Please see issue 4) and show case example to illustrate that fatty hilum was included in the analysis.

7. How were the training and validation cases separated in the analysis, also in Figure 3 and Figure 5?

Reviewer #2: I would like to thank the authors for a well written scientifically sound manuscript addressing an important topic which is being more and more explored in the current era. Hopefully more studies such as this one would help lead the community to perform bigger controlled trials and make the AI softwares vastly available in the clinic in the near future.

6. PLOS authors have the option to publish the peer review history of their article (what does this mean?). If published, this will include your full peer review and any attached files.

Reviewer #1: No

Reviewer #2: No

---

## [Author Response · Author response to Decision Letter 0]

22 Jan 2021

Response to the concerns raised by the reviewers (The First Part) and the academic editor (The Second Part) (quoted in italics and numbered for ease of reference). All changes have been indicated red in the marked-up copy of revised manuscript labeled 'Revised Manuscript with Track Changes copy'.

The First Part (Response to the comments raised by reviewers):

1.Response to Reviewer #1

Response 1: We would like to thank the reviewer for the kind and helpful advice. This was a preliminary study in which we focused on the feasibility of use of radiomic features of axially lymph nodes in preoperative diagnosis of their metastatic status in breast cancer. In the study, we have used the Delong test to compare the overall diagnostic performance of different models. The result showed that the overall diagnostic performance of the combined model and the radiomic model were significantly better than that of the pharmacokinetic model and the conventional model (P<0.05). In clinical setting, it is really hard to demonstrate how the radiomics model may help in every single case in which the reading was wrong by conventional image feature. Following your helpful advice and based on this preliminary result, we will try to design another research with larger sample in order to achieve results that are more applicable in clinical settings.

Response 1: We apologize for not showing the Tables 1 and 2 in the original manuscript due to some technical problems. We have corrected the omission in the revised manuscript. Table 1 in original manuscript shows the texture features in the study. Table 2 shows the diagnostic performance (including the sensitivity and specificity) of four models for detection of metastatic axillary lymph nodes.

Response 1: As suggested, we have added a new table (Table 1 in the revised manuscript) to compare the conventional image features between the malignant and benign groups.

Response 1: Thank you for your insightful comment. In our research, the models constructed by training dataset and their overall performance were tested by the validation dataset. Thus, we cannot show which cases were correctly predicted by radiomics but wrongly diagnosed by conventional reading due to the very nature of radiomic analysis. It is hard to show which cases were correctly predicted by radiomics but wrongly diagnosed by conventional reading.

Response 1: The presence of fatty hilum was evaluated based on T2WI sequences. The MRI acquisition which has been described extensively in our previous study[1] were not specific to the current research. Therefore, we presented only a brief description of the MRI acquisition. We apologize for not clearly stating this before. We have made this clearer in the revised manuscript.

1. Luo HB, Du MY, Liu YY, Wang M, Qing HM, Wen ZP, Xu GH, Zhou P, Ren J: Differentiation between Luminal A and B Molecular Subtypes of Breast Cancer Using Pharmacokinetic Quantitative Parameters with Histogram and Texture Features on Preoperative Dynamic Contrast-Enhanced Magnetic Resonance Imaging. Acad Radiol 2020, 27(3):e35-e44.

Response 1: As suggested, we have added the results and discussion in the revised manuscript to address this issue.

Response 1: The early stage of post-contrast image of DCE-MRI was used for drawing the whole nodal ROI. We are so sorry for not clearly stating this before, and we have made this clearer in the revised manuscript. Besides, the fatty hilum was evaluated on T2WI sequences (illustrated in our response to issue 4).

Response 1: The training and validation cases were separated at a ratio of 7 to 3 in the analysis. We have added this information in the revised manuscript and in figure legends 3 and 5.

2.Response to Reviewer #2

Response 2: Thank you very much for your kind comments. We completely agree that it is important to judge the metastatic status of lymph nodes, especially in patients with breast cancer. In clinical settings, it is really difficult to fulfill this task by conventional image features. Radiomics is a novel and prospective way to assess the tumor heterogeneity which may be the unique characteristic of malignant tumors. This was our motivation to perform this study. However, this was a preliminary research and our results need to be verified in larger multi-center studies. The results demonstrate the feasibility of utilization of radiomic features of axillary lymph nodes for diagnosing their metastatic status in patients with breast cancer.

The Second Part (Response to the points raised by the academic editor):

Response 3: We have revised our manuscript in accordance with the additional requirements of PLOS ONE.

Response 3: As suggested, the manuscript and the revised manuscript have been edited and proofread by the Medjaden Bioscience Limited, a professional medical editing company. A copy of our revised manuscript showing the changes using track changes has been uploaded as a *supporting information* file.

Response 3: As suggested, we have addressed the prompts about the data request in our revised cover letter.

Response 3: We have amended the statement in our revised cover letter: “The authors received no specific funding for this work.”

Response 3: The commercial affiliation did not play any role in our study. We have decided to remove the only author employed by the commercial company (GE Healthcare) from the author list. Therefore, there are no changes in our “Funding Statement” and “Competing Interests Statement” in the revised manuscript.

Response 3: As suggested, we have included the tables as part of the main manuscript in the revised manuscript and have removed the individual table files.

---

## [Decision Letter · Decision Letter 1]

1 Feb 2021

Radiomic features of axillary lymph nodes based on pharmacokinetic modeling DCE-MRI allow preoperative diagnosis of their metastatic status in breast cancer

PONE-D-20-30485R1

Dear Dr. Ren,

We’re pleased to inform you that your manuscript has been judged scientifically suitable for publication and will be formally accepted for publication once it meets all outstanding technical requirements.

Kind regards,

Quan Jiang, Ph,D.

Academic Editor

PLOS ONE

Additional Editor Comments (optional):

Reviewers' comments:

Reviewer's Responses to Questions

**Comments to the Author**

1. If the authors have adequately addressed your comments raised in a previous round of review and you feel that this manuscript is now acceptable for publication, you may indicate that here to bypass the “Comments to the Author” section, enter your conflict of interest statement in the “Confidential to Editor” section, and submit your "Accept" recommendation.

Reviewer #1: All comments have been addressed

2. Is the manuscript technically sound, and do the data support the conclusions?

Reviewer #1: Yes

3. Has the statistical analysis been performed appropriately and rigorously? 

Reviewer #1: Yes

4. Have the authors made all data underlying the findings in their manuscript fully available?

Reviewer #1: Yes

5. Is the manuscript presented in an intelligible fashion and written in standard English?

Reviewer #1: Yes

6. Review Comments to the Author

Reviewer #1: One major suggestion in the previous review is to give per-patient based diagnostic results, using both radiomics and conventional models, but the authors replied that it is not possible. This is a bit puzzling, because the developed model can give a malignancy probability, which can be used to give per-lesion diagnosis. Nonetheless, I also agree that this type of analysis can be best performed in a totally independent testing dataset. The quality of the revision has been further improved for publication.

7. PLOS authors have the option to publish the peer review history of their article (what does this mean?). If published, this will include your full peer review and any attached files.

Reviewer #1: No

---

## [Editor Report · Acceptance letter]

15 Feb 2021

PONE-D-20-30485R1 

Radiomic features of axillary lymph nodes based on pharmacokinetic modeling DCE-MRI allow preoperative diagnosis of their metastatic status in breast cancer 

Dear Dr. Ren:

I'm pleased to inform you that your manuscript has been deemed suitable for publication in PLOS ONE. Congratulations! Your manuscript is now with our production department. 

Kind regards, 

on behalf of

Dr. Quan Jiang 

Academic Editor

PLOS ONE